# Exploring Anxiety and Depression Among Medical Undergraduates in South Africa: A Cross-Sectional Survey

**DOI:** 10.3390/healthcare13060649

**Published:** 2025-03-16

**Authors:** Rajesh Vagiri, Kamogelo Mohlabe, Leny Mailula, Favian Nhubunga, Moitshegi Maepa, Mabitsela Mphasha, Mduduzi Mokoena, Nsovo Mayimele, Varsha Bangalee, Wandisile Grootboom, Letlhogonolo Makhele, Neelaveni Padayachee

**Affiliations:** 1Department of Pharmacy, Faculty of Health Sciences, University of Limpopo, Mankweng 0727, South Africa; mampholwanekamogelo@gmail.com (K.M.); mailulaleny@gmail.com (L.M.); nhubungafa@gmail.com (F.N.); moitsegijay02@gmail.com (M.M.); 2Department of Public Health, Faculty of Health Sciences, University of Limpopo, Mankweng 0727, South Africa; mabitsela.mphasha@ul.ac.za; 3Department of Pathology, School of Medicine, University of Limpopo, Mankweng 0727, South Africa; mduduzi.mokoena@ul.ac.za; 4Department of Pharmaceutical Sciences, Faculty of Science, Tshwane University of Technology, Pretoria 0001, South Africa; mayimelenn@tut.ac.za; 5Discipline of Pharmaceutical Sciences, School of Health Sciences, University of KwaZulu-Natal, University Road, Durban 4000, South Africa; bangalee@ukzn.ac.za; 6Medical School, Nelson Mandela University, Uitenhage Road, Port Elizabeth 6031, South Africa; wandisile.grootboom@mandela.ac.za; 7School of Pharmacy, Sefako Makgatho Health Sciences University, Molotlegi St., Ga-Rankuwa, Pretoria 0208, South Africa; letlhogonolo.makhele@smu.ac.za; 8Department of Pharmacy and Pharmacology, School of Therapeutic Sciences, Faculty of Health Sciences, University of Witwatersrand, Johannesburg 2193, South Africa; neelaveni.padayachee@wits.ac.za

**Keywords:** anxiety, depression, well-being, mental health, severity

## Abstract

**Background/Objectives:** Globally, there has been an increase in the prevalence of anxiety and depression among university students, and medical students are no exception. Medical students are especially susceptible to these mental health challenges, primarily due to multifaceted stressors, which can significantly impact their academic achievements and future career. There is a pressing need for comprehensive research that not only investigates the prevalence of anxiety and depression among medical students but also explores strategies for developing effective mental health interventions and support systems that can enhance the well-being of medical students. Therefore, this study aimed to identify the prevalence and severity of anxiety and depression among medical students at a university in South Africa, evaluating the association of socio-demographic, student, and clinical variables with total general anxiety disorder (GAD-7) and patient health questionnaire (PHQ-9) scores. **Methods:** A survey-based quantitative cross-sectional study was conducted with 208 medical undergraduate students at a South African university. Participants who provided written consent completed GAD-7 and PHQ-9 questionnaires together with socio-demographic, student, and clinical variable information. The relationship between socio-demographic, student, and clinical variables and total GAD-7 and PHQ-9 scores was determined using the Mann–Whitney U test and Kruskal–Wallis H test. Correlation analysis was used to establish the relationship between total anxiety and depression scores. The threshold for statistical significance was set at *p* ≤ 0.05. **Results:** More than half of the participants were female (n = 130; 62.5%), single (n = 123; 59.1%), and belonged to the *Pedi* ethnic group. A majority of the students were Christian (n = 183; 88.0%), received a bursary (n = 183; 88.0%), and had a rural background (n = 155; 74.5%). However, a small percentage of students reported a history of psychiatric and chronic illnesses (n = 26; 12.5%) and previously received professional psychological support (n = 38; 18.3%). In this study, 38% (n = 79) of the participants reported GAD and 67.8% (n = 141) reported symptoms of depression. Significant associations (*p* <0.05) were observed between variables such as year of study, repeating a module, and history of psychiatric illness with total GAD-7 and PHQ-9 scores. Correlation analysis revealed a moderate positive correlation (*r*_s_ = 0.400, *df* = 206, *p* < 0.001) between total GAD-7 and PHQ-9 scores. **Conclusions:** This study identified a high level of depression and anxiety among medical students and found a positive correlation between anxiety and depression scores. Addressing these mental health challenges is crucial not only for the well-being of the students but also for the future of healthcare, as the mental health of medical professionals directly impacts patient care.

## 1. Introduction

Mental health conditions, particularly anxiety and depression, have emerged as significant public health concerns globally [1,2]. The World Health Organization (WHO) has reported that anxiety and depression are among the leading causes of disability worldwide, affecting millions of individuals across diverse demographics and socio-economic backgrounds [3]. Anxiety disorders encompass a range of conditions characterised by excessive fear or worry, which can manifest in increased heart rate, sweating, and restlessness [4]. Depression, on the other hand, is marked by persistent feelings of sadness, hopelessness, and a lack of interest or pleasure in activities once enjoyed [3]. These two conditions often co-occur, with studies indicating that individuals with anxiety are at a higher risk of developing depression and vice versa [5,6].

The mental health of university students, particularly in terms of anxiety and depression, has become a pressing issue in recent years, exacerbated by various stressors inherent in the academic environment [7]. Research suggests that university students are particularly vulnerable to mental health challenges due to a combination of academic pressures, social expectations, and the transition to adulthood and university [8,9]. Additionally, medical students, compared to other categories of health professions, face particularly rigorous academic demands, including long hours of study and high-stakes assessments, which may contribute to their heightened mental health challenges [10].

### 1.1. Background

The interplay between academic stressors and mental health is complex, with many students experiencing a decline in academic performance [11]. Al Ani et al. observed that students with superior mental health generally achieved higher academic performance, underscoring the notion that mental well-being is essential for academic achievement. Stress, anxiety, and depression adversely affect sleep quality and academic performance, suggesting that mental health concerns can result in diminished academic achievements [12]. Anxiety and depression can hinder decision-making processes, resulting in less effective clinical judgements. Students who are preoccupied with mental health challenges may experience a diminished capacity for critical thinking and sound decision-making in clinical environments [13]. This is especially alarming in high-stakes settings when precise decision-making is critical for patient safety and care quality. Medical students suffering from anxiety and depression may find it challenging to establish sympathetic relationships with patients, which is essential for good healthcare provision. Mental health disorders can result in burnout, therefore reducing the quality of care delivered to patients [14]. Inadequate management of mental health among medical students can lead to diminished patient satisfaction and adverse health outcomes [15]. The enduring consequences of unaddressed mental health issues in medical students can impact their career viability. Elevated stress and burnout can result in premature career departures, diminished job satisfaction, and heightened attrition rates within the medical field [14].

The prevalence rates of anxiety and depression have varied widely across different regions and educational contexts. A systematic review and meta-analysis conducted by Li et al. revealed that the global prevalence of anxiety among medical students is approximately 33.8%, while the prevalence of depression is 28.0% [15]. A study conducted by Shao et al. reported a prevalence of depression at 57.5% among Chinese medical students [16]. Similarly, an Iranian study found that 38% of their medical students reported experiencing anxiety [17]. The prevalence of anxiety and depression is not uniform across all medical students; factors such as year of study, gender, and individual coping mechanisms also play a significant role. First-year medical students often report higher levels of anxiety compared to their senior counterparts, as they navigate the challenges of adapting to medical school [18,19]. Additionally, studies have shown that female medical students tend to experience higher rates of anxiety and depression than their male peers, which may be linked to societal expectations and gender-related stressors [20,21]. The consequences of untreated anxiety and depression among medical students can be severe, impacting not only their academic performance but also their overall well-being. Research has demonstrated that students with higher levels of anxiety and depression are more likely to experience poor sleep quality, decreased academic performance, and increased risk of burnout [12,22].

The prevalence of anxiety and depression among medical students in South Africa is a growing concern, reflecting a broader trend observed globally within this demographic. Research indicates that a notable proportion of medical students grapple with various mental health issues including anxiety and depression [23]. A study conducted at the University of Cape Town found that approximately 27.2% of medical students reported experiencing depressive symptoms, which is significantly higher than in their age-matched peer group in the general population [24].

The factors contributing to anxiety and depression among medical students in South Africa are multifaceted. Academic pressure, financial stress, and the transition to a demanding professional environment are significant contributors. A study by Chigerwe et al. noted that cultural differences and race also play a role in shaping the mental health experiences of medical students, indicating that these issues are not solely academic but are influenced by broader societal factors [25]. Additionally, the high levels of stress associated with medical training can lead to burnout, which is closely linked to anxiety and depression [26]. In response to the mental health crisis among medical students, South African universities are beginning to implement support systems and resources aimed at promoting mental well-being. However, there remains a significant gap in mental health resources available to students, particularly in South Africa, where access to psychological support is limited [27,28].

Multiple socio-demographic, academic, and clinical factors may influence students’ experiences and coping strategies in the rigorous context of medical education and substantially affect the prevalence and severity of anxiety and depression in medical students. Age and year of study (academic level) are deemed significant determinants, as younger medical students frequently exhibit elevated levels of anxiety and depression relative to their older peers. First-year students often encounter increased anxiety stemming from the transition to medical school and the accompanying academic expectations [29,30]. A study indicated that the prevalence of anxiety symptoms was 30.8% among first-year students, decreasing to 9% in sixth-year students [30]. This trend may be ascribed to the difficulties of transition and heightened academic expectations encountered by newcomers, which can result in feelings of inadequacy and stress [29]. As students advance in their studies, they generally grow more familiar with the rigours of medical education, potentially mitigating some anxiety and depression symptoms [31].

Gender disparities significantly influence mental health outcomes among medical students. Studies have consistently indicated that female students are more prone to anxiety and depression compared to their male counterparts [32,33]. This disparity may be ascribed to multiple variables, such as cultural expectations, self-imposed stress, and variations in coping mechanisms. Female students frequently exhibit elevated levels of perfectionism and self-expectation, which may intensify feelings of inadequacy and anxiety [33]. Cultural attitudes toward mental health can affect the ways in which male and female students articulate and pursue assistance for their mental health concerns [34]. The status of relationships significantly influences the mental health of medical students. Individuals in supportive relationships may exhibit reduced anxiety and depression levels owing to the emotional support offered by their partners [35]. In contrast, students who are either single or facing relationship challenges may experience heightened feelings of loneliness and isolation, potentially worsening their mental health issues [36]. The emotional strain of sustaining relationships while managing academic obligations can exacerbate stress, especially for students who find it challenging to reconcile their personal and academic life [37].

Research indicates that medical students from rural backgrounds tend to experience higher levels of anxiety and depression compared to their urban counterparts. Students from rural areas face unique challenges such as limited access to mental health resources, social isolation, and economic hardships that can exacerbate mental health issues [38]. A systematic review by Mao et al. highlighted that those medical students who lived in rural areas exhibited increased symptoms of anxiety, particularly those who were ethnic minorities or had poor interpersonal relationships [39]. Similarly, a study conducted in Nigeria noted that students from public institutions, which are often located in the rural areas, exhibited higher levels of anxiety and depression compared to those in private institutions that are based in the urban areas [40].

The interplay between financial strain and academic demands might exacerbate mental health issues. This relationship suggests that financial issues can exacerbate the elevated stress levels inherent in medical school, resulting in heightened anxiety and depression [41]. Hwang et al. highlighted that financial strain and debt burden are prevalent sources of stress for students, potentially resulting in burnout and depression, which may adversely affect their future roles as healthcare professionals [42]. Students from economically disadvantaged backgrounds may experience heightened anxiety and depression as a result of financial burdens, such as tuition and living costs [37,43]. A significant negative causal link exists between household income and anxiety, revealing that those with elevated household incomes displayed reduced levels of anxiety [44]. The burden of managing monthly costs can intensify mental health challenges among students [45]. Obtaining a bursary might mitigate financial strain, as sufficient financial assistance may serve as a protective factor for mental health, potentially diminishing anxiety and depression [46].

Medical students who are parents frequently encounter heightened anxiety and depression stemming from the distinct problems of reconciling academic obligations with parenting duties. The necessity of efficiently managing time to fulfil educational and familial responsibilities might overwhelm students, leading to heightened mental health issues [47].

Living arrangement, including whether students cohabit with parents or relatives or live alone, can influence their mental health. Students residing with both parents generally exhibit reduced levels of anxiety and depression, as parental emotional support correlates with a significant reduction in these symptoms among students, underscoring the need for a loving and supportive home environment [48]. On the other hand, the nature of accommodation, whether on-campus or off-campus, also influences students’ mental health. On-campus housing may offer enhanced social support and interaction opportunities, perhaps alleviating feelings of loneliness and anxiety [49]. This indicates that the campus environment may provide a protective role for mental health, especially during challenging times [50].

Studies demonstrate that spirituality and religious practices can operate as protective factors against anxiety and depression, enhancing resilience and coping strategies among medical students [51,52]. In numerous societies, religious beliefs offer a framework for comprehending life’s obstacles, particularly the stressors linked to medical training. This cultural dimension is essential, as it influences students’ perceptions of mental health and their readiness to seek assistance [52]. Students who actively participate in religious traditions may discover comfort and support within their groups, perhaps reducing feelings of loneliness and anxiety [52].

Student variables such as failing or repeating a module and accommodation may influence anxiety and depression levels among medical students. Academic performance is a significant predictor of mental health outcomes in medical students. Encountering failure or the requirement to retake modules can lead to increased stress, anxiety, and feelings of inadequacy. Pressure to succeed in a competitive environment can exacerbate feelings of self-doubt and hopelessness [53,54]. The fear of failure may create a harmful cycle where performance anxiety leads to reduced academic success, hence worsening mental health issues [55].

Clinical variables, such as a history of psychiatric disorders, the presence of chronic illnesses, previous trauma exposure, and engagement in psychological therapy, are acknowledged risk factors for anxiety and depression in medical students. Medical students with a history of psychiatric disorders are at an increased risk of developing anxiety and depression during their studies, since prior mental health concerns may predispose them to elevated psychological distress in the rigorous context of medical education [56]. The stigma associated with mental health concerns may dissuade medical students from seeking assistance, thereby intensifying their anxiety and depression [57]. Mental and physical illnesses can reduce work effectiveness and elevate absenteeism, consequently resulting in heightened anxiety and depression among medical students [9]. Students with chronic diseases may encounter weariness, pain, or other symptoms that impede their capacity to fully participate in their academics. This can establish a cycle of stress and anxiety, wherein scholastic difficulties further aggravate mental health concerns [58].

Prior trauma history of medical students correlates with an increased risk of post-traumatic stress disorder (PTSD) and major depressive episodes, suggesting that those who have undergone multiple traumatic experiences are more prone to complicated mental health issues [59]. The cumulative impact of trauma exposure may result in heightened psychological distress during the medical training, rendering individuals more vulnerable to anxiety and depression, significantly affecting their mental health outcomes [60]. Although psychological support aims to mitigate mental health difficulties, several variables may inadvertently exacerbate anxiety and depression in this demographic. The psychological help offered to medical students is frequently short-term and deficient in thorough follow-up. Temporary solutions may insufficiently resolve the fundamental difficulties, thereby exacerbating feelings of anxiety and depression when students recognise that their problems endure despite seeking assistance [61]. The stigma surrounding mental health concerns within medical education may dissuade students from seeking assistance or induce feelings of guilt over their mental health challenges. This stigma can establish a loop in which students feel obligated to conceal their difficulties, resulting in heightened anxiety and sadness when psychological treatment fails to address their needs [57].

Most studies conducted in South Africa targeted primarily on assessing mental health conditions in general with limited focus on anxiety or depression together. Although a few studies were conducted in urban areas such as Cape Town, to our knowledge no study focused on assessing the prevalence and severity of anxiety and depression among medical students at a rural university.

### 1.2. Purpose

Given the growing body of evidence highlighting the burden of mental health disorders among medical students globally and locally, there is an urgent need for research that comprehensively evaluates the prevalence and severity of anxiety and depression, particularly within underrepresented contexts such as rural universities in South Africa. Existing research has largely overlooked these settings, despite the heightened vulnerabilities faced by rural students. Given these insights, there is an urgent need for comprehensive research that measures the prevalence of mental health issues among medical students and that investigates the underlying factors contributing to these problems. This research is essential for developing effective mental health interventions and support systems that enhance the well-being of medical students, hence improving their academic performance and patient care outcomes.

Therefore, this study aimed to assess the prevalence of anxiety and depression among medical undergraduates at a rural university in South Africa with the following specific research questions.

What is the severity of anxiety and depression among medical undergraduate students at a rural university in South Africa?Which socio-demographic, student, and clinical characteristics are associated with anxiety and depression levels?Is there a significant correlation between anxiety and depression levels among these students?

## 2. Materials and Methods

This research was executed and documented in compliance with the Strengthening the Reporting of Observational Studies in Epidemiology (STROBE) (requirements for cross-sectional studies. The STROBE checklist was employed to guarantee thorough and transparent reporting of all critical elements of the investigation.

### 2.1. Study Design and Sampling

This study was descriptive and cross-sectional, based on survey data. The research was performed with 208 medical students at a university in South Africa. Total purposive sampling was utilised to choose the study population.

### 2.2. Data Collection Tools

Respondents of this study completed three questionnaires, namely GAD-7, PHQ-9, and a survey of socio-demographic, clinical, and student characteristics.

Generalised Anxiety Disorder (GAD-7) Scale: The GAD-7 is specifically intended to evaluate symptoms of GAD, providing a swift and effective tool for clinicians to identify individuals who may need additional assessment or care. The examination consists of seven items that measure the frequency of anxiety symptoms experienced in the prior two weeks, with a total score ranging from 0 to 21. The responses for all items included ‘not all’ (0), ‘many days’ (1), ‘more than half the days’ (2), and ‘almost every day’ (3) [62].

Patient Health Questionnaire (PHQ-9): The PHQ-9 evaluates the existence and intensity of depressive symptoms according to the Diagnostic and Statistical Manual of Mental Disorders, Fifth Edition (DSM-V) criteria for major depressive disorder. The evaluation comprises nine elements that represent the fundamental symptoms of depression, including mood, interest, and energy levels, with a total score ranging from 0 to 27 [63,64]. The participants were instructed to evaluate each item in terms of the frequency of their experiences with the signs and symptoms of depression over the past two weeks. All items received replies of ‘not all’ (0), several days’ (1), ‘more than half the days’ (2), and ‘nearly every day’ (3).

The questionnaire on socio-demographic, student, and clinical variables comprised three components. The initial portion consisted of socio-demographic data, including gender, age, and ethnicity. The second portion included student attributes such as academic year, bursary details, and monthly stipend. The final segment encompassed clinical criteria such as a history of psychiatric disease and the presence of chronic disorders, among others.

### 2.3. Data Collection

Prior to data collection, a priori power analysis was performed using G*Power version 3.1.9.7 to ascertain the minimal sample size necessary for identifying a medium effect size (r = 0.30) with a two-tailed test, an α of 0.05, and a power of 0.80. This analysis revealed that 85 participants were required for this study. Our final sample of 208 participants surpassed this criterion, guaranteeing enough power.

The data were gathered during a period of four weeks in October 2024. Participants recruited in this study were aged 18 and older, enrolled in 2024, and had given written agreement to participate. Individuals who chose not to participate were omitted from the study. The researchers provided a study information sheet detailing the aims, objectives, and purpose of the investigation. Students who had provided informed consent to participate in this study were given the questionnaires to complete. The participants completed the questionnaires in a secluded area on the university campus and were allotted sufficient time for completion.

### 2.4. Reliability and Validity

The GAD-7 and PHQ-9 questionnaires exhibited strong test–retest reliability and good internal consistency. Both questionnaires have been translated into multiple languages and are extensively used in clinical practice and research globally [65,66,67]. The PHQ-9 demonstrated exceptional internal consistency, with Cronbach’s alpha values generally surpassing 0.80, signifying that the items within the questionnaire assess the same underlying concept [68,69]. GAD-7 also demonstrated strong reliability, with Cronbach’s alpha values between 0.89 and 0.92 across several populations, indicating its effectiveness as a tool for evaluating GAD [62].

This study employed established data collection protocols, and all the completed questionnaires were validated for completeness. This study adhered rigorously to the predetermined inclusion and exclusion criteria, hence avoiding selection bias. The research team underwent comprehensive training to reduce information bias. Response bias was mitigated by the self-administration of the data collection instruments. Language and translation bias were eliminated by administering all the questionnaires only in English.

### 2.5. Ethical Considerations

This study obtained ethical approval from the institutional research ethics committee. Authorisation to conduct the study was secured from the university registrar, and approval from the gatekeeper was gained from the School of Healthcare Sciences. The researchers upheld confidentiality by abstaining from the collection of any personal information from participants, including student identification numbers, national identification numbers, and mobile phone numbers. Before data collection, all participants received a study information sheet detailing the aim, objectives, risks, and benefits of the study. Participation was wholly voluntary, and written informed consent was secured from each participant prior to data collection. Participants were notified of their ability to withdraw from the study at any moment without justification. Participants were enlisted via class announcements. No financial or material incentives were offered for participation. The lack of incentives was explicitly conveyed to prospective volunteers to prevent any undue influence. Confidentiality was additionally ensured by securely storing all data and tools utilised in this investigation, with access limited strictly to the research team. The participants’ privacy was maintained by using participant codes linked to their academic year. Participants were notified that their data would be utilised exclusively for academic research, with the findings published in scientific journals and presented at conferences. They were guaranteed that no person would be recognisable in any reports or publications resulting from the study. The researchers mitigated harm in this study by ensuring minimum risk, with counsellors from the university’s Counselling and Development Centre available on standby if necessary. This study complied with all ethical practices and requirements.

### 2.6. Data Analysis

Data pertaining to socio-demographic, student, and clinical variables, along with the GAD-7 and PHQ-9 questionnaires, were documented in Microsoft Office Excel and analysed utilising Statistical Package for the Social Sciences (SPSS) version 29.0. Before analysis, the dataset was meticulously scrutinised for missing values utilising descriptive statistics and frequency assessments in SPSS. No missing data were detected among all variables incorporated in the study. This was presumably enabled by real-time monitoring and verification to assure data completeness. Consequently, no imputation methods or techniques for managing missing data were required.

Data concerning socio-demographic, student, and clinical variables were presented as frequencies and percentages. Normality tests pertaining to the data were conducted using a one-sample Kolmogorov–Smirnov test. The total GAD-7 scores were calculated by summing all replies for each item, resulting in a score ranging from 0 to 21. The total GAD-7 scores were classified as ‘no symptoms’ (0–4), ‘mild’ (5–9), ‘moderately severe’ (10–14), and ‘severe’ (≥15) to evaluate anxiety severity. To evaluate the severity of depression, the PHQ-9 responses were aggregated to yield a total score ranging from 0 to 27, categorised as follows: 0–4 (‘none’), 5–9 (‘mild’), 10–14 (‘moderate’), 15–19 (‘moderately severe’), and 20 or above (‘severe’). A score of 10 or above on both scales generally signifies moderate to severe anxiety and depression.

Associations of socio-demographic, student, and clinical variables with total GAD-7 and PHQ-9 scores were identified utilising the Mann –Whitney U test and Kruskal–Wallis H test when appropriate. A correlation analysis was performed to ascertain the degree and direction of the connection between total GAD-7 and PHQ-9 scores. The threshold for statistical significance in this investigation was established at *p* < 0.05.

Considering the cross-sectional and self-reported design of our investigation, procedural and statistical strategies were implemented to mitigate common method bias. Harman’s single-factor test was employed to identify common method variance. The unrotated factor analysis indicated that the first component explained 33.53% of the variation for the GAD-7 scale and 28.11% for the PHQ-9 scale, which are well below the accepted 50% threshold variance, suggesting that common method bias was insignificant in our study.

## 3. Results

Out of 366 medical undergraduates registered in 2024, a total of 208 students consented in writing to participate in this study, resulting in a response rate of 57%.

### 3.1. Socio-Demographic, Student, and Clinical Variables

In this study, the majority of the participants were Christian (n = 183; 88.0%), received a government bursary (n = 183; 88.0%), resided on campus (n = 162; 77.9%), and had a rural background (n = 155; 74.5%). More than half of the participants were female (n = 130; 62.5%), single (n = 123; 59.1%), belonged to the *Pedi* ethnic group (n = 107; 51.4%), and survived on a monthly allowance of 501–2000 South African Rands (n = 129; 62.0%). A small percentage of students reported a history of psychiatric and chronic illnesses (n = 26; 12.5%) and had previously received professional psychological support (n = 38; 18.3%). Table 1 shows the socio-demographic, student, and clinical variable data of the study population

### 3.2. Reliability of GAD-7 and PHQ-9 Scales

A Cronbach’s alpha analysis was conducted to evaluate the internal consistency of the seven-item GAD and nine-item PHQ scales. For the GAD-7 scale, the internal consistency was determined to be 0.67, signifying an acceptable level of reliability. The item–total correlations varied from 0.61 to 0.65, indicating that all items positively contributed to the overall reliability.

The PHQ-9 demonstrated an internal consistency of 0.67, indicating a satisfactory level of reliability. Item–total correlations ranged from 0.61 to 0.66, implying that each item played a positive role in maintaining overall reliability.

### 3.3. GAD-7 Responses and Severity of Anxiety Among Medical Students (n = 208)

A majority of the subjects indicated experiencing anxious symptoms for several days, often on a daily basis. Moderate mean scores were documented varying from 1.13 to 1.36. Question 4 (trouble relaxing) and question 2 (not being able to stop or control worrying) received higher mean scores of 1.36 and 1.32, respectively. Table 2 presents the responses to GAD-7 scale.

In this study, students who reported moderate to severe levels (scores exceeding 10) were deemed to exhibit symptoms of GAD. Elevated levels (moderate and severe) of GAD were observed in 38% (n = 79) of the participants, while over half (51.9%; n = 108) of the medical students reported mild symptoms of GAD (Figure 1).

### 3.4. PHQ-9 Responses and Severity of Depression Among Medical Students (n = 208)

The mean scores for the responses to the PHQ-9 ranged from 0.64 to 1.54, with most individuals reporting feelings of depression occurring from several days to more than half the days. A more favourable response was noticed for Question 3 (moving or speaking so slowly that other people could have noticed, or the opposite—being so fidgety or restless that you have been moving around a lot more than usual), which elicited a predominant response of ‘not at all’. A majority of participants experienced difficulties in trouble falling or staying asleep or sleeping too much (n = 182; 87.5%) for several days to nearly every day (Question 3), leading to a higher mean score of 1.54 (Table 3).

Students exhibiting moderate, moderately severe, and severe depression were classified as having depressive symptoms. Over two-thirds of the participants (n = 141; 67.8%) indicated experiencing symptoms of depression (PHQ score >10). More than half (52.9%; n = 110) of the participants reported experiencing mild symptoms of depression (Figure 2).

### 3.5. Association of Socio-Demographic, Student, and Clinical Variables with Total GAD-7 and PHQ-9 Scores

This study identified a strong association (*p* <0.05) between variables such as year of study, repeating a module, and history of psychiatric illness with total GAD-7 and PHQ-9 scores. A significant association (*p* = 0.01) between ethnicity and anxiety scores was observed, with the ‘*Venda*’ ethnic group reporting elevated anxiety levels (10.07) compared to other ethnic groups. Age significantly (*p* = 0.00) affected the experience of depressed symptoms. Students aged 20–21 exhibited a heightened level of depression (11.90) relative to other age groups. A notable association (*p* = 0.00) was identified between household income and anxiety levels. Household type had a significant influence on the levels of depression. Students who cohabited with a single parent reported high levels of depression (mean: 11.72) compared to the students who lived with both parents or relatives. Unexpectedly, participants with a household income exceeding ZAR 20,000 had marginally elevated anxiety levels (9.37) compared to those in the other household income brackets (Table 4).

### 3.6. Correlation Between Total GAD-7 and PHQ-9 Scores

This study identified a moderate positive correlation (*r*_s_ = 0.400, *df* = 206, *p* <0.001) between total GAD-7 and PHQ-9 scores reported by the medical students (Table 5).

## 4. Discussion

The study results indicate that 38% (n = 79) of the medical students reported GAD, whereas 67.8% (n = 141) reported the symptoms of depression. Our study findings are consistent with a broader trend of elevated mental health issues in this demographic, underscoring the mental health challenges faced by medical students worldwide [70]. A systematic review and meta-analysis revealed that the prevalence of depression among medical students is approximately 27.2% [54]. However, our study finding of 67.8% signifies a markedly higher rate of depressive symptoms. This discrepancy may be ascribed to several reasons, including the specific population studied, the instruments used for assessment, and the contextual pressures faced by students in diverse settings. Furthermore, a study conducted at Alexandria University found that 57.9% of medical students experienced depression, closely correlating with our findings [71]. Similarly, research from India reported a prevalence of depression at 71.25% among medical students, highlighting the significant prevalence of mental health issues in this population [72]. The prevalence (38%) of GAD determined in our study is higher than the global average of 4.8% to 10.9% as reported in a meta-analysis by Quek et al. (2019) [34]. However, other studies have reported elevated rates of anxiety, including 43.9% at Alexandria University [71] and 19.9% in a Chinese study, reflecting a significant concern for mental health among medical students [73]. The variation in anxiety prevalence may be attributed to different evaluation tools and the specific stressors encountered by students in distinct educational settings. Furthermore, the cumulative impact of academic pressures, social isolation, and the rigorous demands of medical education have been observed to exacerbate mental health issues among medical students [74]. The disparity in reported prevalence rates across different studies highlights the need for ongoing research and tailored mental health support strategies within medical education.

Fifth-year medical students in our study reported elevated levels of anxiety and depression. Several studies have reported that students in clinical years have heightened levels of anxiety and depression relative to their preclinical counterparts. Mahgoub et al. found that fifth-year students exhibited higher levels of anxiety and depression than students in earlier years, suggesting that the impending graduation and clinical obligations may intensify mental health issues [27]. In contrast, Hassnain et al. observed that students transitioning from preclinical to clinical years reported significant depressive symptoms, with 57.7% of third-year students experiencing moderate to severe depression [75]. This is echoed by Jafari et al., who suggested that the substantial workload and adaptation to clinical environments contribute to worsening mental health as students advance in their studies [76]. Numerous studies suggest that first-year students frequently experience elevated levels of anxiety and depression; however, there is also evidence that clinical years may pose unique challenges that may lead to increased mental health issues. The variability in findings indicates that the relationship between the year of study and mental health is influenced by a multitude of factors, including academic workload, social support, and individual resilience, and underscores the need for tailored mental health support throughout students’ education.

The association identified in our study between young age and depression among medical students has been a subject of considerable research, indicating that younger medical students, especially those in their first and second years, are at an increased risk of depression compared to their older peers [77]. This corresponds with findings from Roh et al., which demonstrated that the prevalence of depression among medical students was notably higher than that of the general population, particularly in the younger age group [78]. The stress associated with transitioning to medical school, coupled with the rigorous and demanding curriculum, might intensify feelings of inadequacy and hopelessness, resulting in increased depressive symptoms [34]. This suggests that the developmental phase of younger students may contribute to their vulnerability to mental health issues, as they may not possess the coping mechanisms that older students have developed over time [79]. This highlights the importance of addressing mental health concerns early in medical education to foster both personal and academic success.

Medical students who cohabited with both parents reported significantly lower levels of depression in comparison to those living with a single parent or with relatives. Cohabitation with both parents can offer emotional support, stability, and resources that may alleviate the risks of anxiety and depression, especially in high-stress environments such as medical school. Studies suggest that students not residing with biological parents frequently have elevated levels of depressive symptoms [80]. A study examining Filipino university students revealed that individuals not living with both parents exhibited more severe depressive symptoms, suggesting that parental presence is vital for emotional well-being [81]. This aligns with findings from a systematic review that emphasised the significance of parental support in safeguarding adolescents against depression [82]. The absence of both parents may result in feelings of isolation and insufficient social support, which are notable risk factors for the onset of anxiety and depression [83]. Furthermore, the financial and emotional support provided by cohabiting parents can alleviate stressors associated with medical education [81]. Additionally, the quality of parental relationships is crucial. Chang et al. indicated that the parent–child relationship substantially affects the mental health of medical students, with positive interactions associated with reduced anxiety and depression. This suggests that the presence of both parents and the quality of familial interactions might influence students’ mental health outcomes [82]. Further research is necessary to explore these dynamics in greater depth, particularly in diverse cultural contexts.

Research consistently demonstrates that individuals with a history of trauma exposure are at an increased risk for developing anxiety and depressive disorders. In our research, medical students with a history of trauma reported high levels of anxiety. A study by Bahar revealed a positive correlation between childhood trauma and the manifestation of anxiety and depressive symptoms in adulthood, highlighting the enduring impact of early traumatic events on mental health [84]. The ramifications for medical students are especially pertinent, as they frequently face high-stress environments that can exacerbate pre-existing vulnerabilities related to trauma. This aligns with findings from Dunn et al., which revealed that individuals exposed to trauma, irrespective of the timing of exposure, exhibited significantly higher levels of depression and PTSD symptoms in comparison to those without such exposure [85]. Furthermore, Larson et al. emphasised that the frequency of trauma exposure significantly influences the emergence of mental health disorders, especially within academic settings [86]. This suggests that medical students with a history of trauma may experience compounded stressors associated with their rigorous training, resulting in heightened levels of anxiety. A study by Stewart et al. found that adolescents exposed to multiple traumatic events reported elevated scores on measures of PTSD, anxiety, and depression, indicating that cumulative trauma exposure adversely impacts mental health [87]. The relationship between trauma exposure and mental health outcomes is further supported by research indicating that trauma can lead to maladaptive coping mechanisms, which may exacerbate symptoms of anxiety and depression [88]. This is particularly concerning for medical students who may be contending with difficulties of their education while grappling with unresolved trauma. The findings suggest that medical schools should implement comprehensive mental health support systems to address the unique challenges faced by students with trauma experiences.

Higher household income has a notable influence on anxiety levels among medical students. Interestingly, in our study, students who reported higher household income (ZAR > 20,000) reported heightened anxiety. Research suggests that stress, anxiety, and depression are not directly associated with the household income. However, the psychological impact of financial instability could lead to increased anxiety levels [45]. This indicates that the relationship between income and anxiety is influenced by various factors, including the overall mental health landscape and individual coping mechanisms.

Our research revealed that the repetition of a module in medical school might substantially affect anxiety and depression levels in medical students. Studies demonstrate that the stress linked to academic achievement is a major factor in the development of anxiety and depression among students [89,90]. The stigma associated with repeating a module may induce emotions of inadequacy and self-doubt, exacerbating mental health concerns [91]. Students required to retake a module may encounter a sense of failure, perhaps resulting in hopelessness and depression [92]. The reduction in self-esteem might initiate a detrimental loop, wherein diminished confidence results in inferior academic performance, thereby exacerbating anxiety and depression symptoms [93]. Individuals possessing efficient coping mechanisms and robust support networks may be more adept at managing the difficulties related to scholastic disappointments. In contrast, students without these resources may endure increased anxiety and despair [43,94]. This highlights the necessity of proactively addressing the mental health requirements of medical students, especially those at risk of module repetition.

Our study indicates that medical students of ‘*Venda*’ ethnicity exhibit elevated anxiety levels. This finding lacks a straightforward explanation but can be ascribed to a confluence of cultural, social, and academic variables. Comprehending these variables is essential for formulating focused solutions to assist this demographic.

Our study established an association between a history of psychiatric illness and the prevalence of anxiety and depression among medical students, underscoring the influence of pre-existing mental health conditions on the psychological well-being of students in high-stress academic environments such as medical school. Ali et al. found that students with a history of psychiatric illness had elevated levels of psychological distress, suggesting that these individuals are more susceptible to external stressors [95]. Karbownik et al. reported that medical students with psychiatric diagnoses exhibited more severe depressive and anxiety symptoms than their psychiatrically healthy peers, with a significant proportion of these students presenting with at least mild symptoms of both conditions [96]. This suggests that a history of psychiatric illness not only predisposes students to mental health challenges but also influences the severity of these symptoms. The recurrence of psychiatric symptoms in these individuals highlights the necessity for targeted mental health interventions within medical education. Moreover, the stigma associated with mental illness can influence how medical students with a history of psychiatric illness perceive their condition and seek help. Sampogna et al. found that students with a prior psychiatric history exhibited distinct attitudes towards seeking help in contrast to those without such a history, indicating that personal experiences with mental illness may influence help-seeking behaviours [97]. This highlights the importance of creating supportive environments that encourage students to seek help without fear of stigma.

The discussion of the implications stemming from the findings of anxiety and depression among medical students highlights a critical necessity for further research, targeted interventions, and adjustments within medical education. Considering the multifaceted nature of mental health disorders, which are intricately linked to academic pressure, financial stress, and the challenging transition into healthcare professions, strategic initiatives are crucial for alleviating these challenges.

An important area for future research lies in the exploration of demographic disparities affecting mental health in medical students. Research suggests that first-year medical students experience elevated anxiety levels relative to their older counterparts due to the challenges of initial adjustments and demands [77,98]. Future research should delve into the factors influencing these variations, considering both the year of study and gender differences, as research consistently indicates that female medical students have elevated levels of anxiety [99,100]. Tailoring mental health interventions to meet these demographic particulars can improve their effectiveness and accessibility.

Furthermore, it is imperative for future research to thoroughly examine the psychological effects of academic pressures on medical students. Studies indicate a significant association between academic stress and increased rates of burnout, especially within medical training [98,101]. Longitudinal studies may provide insights into the evolution of academic stresses during medical education and their consequent impact on students’ mental health information, which can guide the development of interventions designed to enhance student resilience.

Establishing organised support systems within medical schools is essential. Mentorship programs tailored for first-year students can be particularly advantageous, offering support during their transition into medical education [98,100,101]. Findings indicate that students participating in peer-supported activities have superior stress management skills [102,103]. Implementing programs aimed at improving adaptive coping strategies such as mindfulness and stress management training can substantially reduce the mental health burden experienced by this group [101].

The existing constraints on mental health support in educational environments, particularly in contexts such as South Africa, require collaborations between medical schools and mental health institutions to enhance the availability and accessibility of resources [104]. Comprehensive support frameworks must incorporate not only psychological counselling but also workshops encompassing stress management techniques. Employing these strategies can equip students with essential skills to adeptly manage their professional expectations while preserving their mental wellness [98].

Ultimately, fostering an atmosphere that reduces the stigma associated with mental health at medical schools is essential [99,100]). Awareness programs aimed at normalising discussions about mental health issues can enable students to seek assistance when necessary. Research indicates that openly discussing mental health helps cultivate a sense of community and support among medical students, thus enhancing general well-being [99,100].

## 5. Limitations

In assessing the limitations of our research on the mental health of medical students, certain aspects must be considered that could have influenced the generalisability and interpretation of our results.

Our study encompassed a relatively small and homogeneous cohort of medical students from a single race; thus, the findings may not accurately reflect the wider community of medical students across other institutions or areas. The self-reported assessments of anxiety and depression may have introduced bias. Participants may have inaccurately reported their symptoms owing to social desirability bias or insufficient awareness of their mental health condition. While this study reports associations between numerous socio-demographic, student, and clinical characteristics with total GAD-7 and PHQ-9 scores, the cross-sectional nature of the data precludes the determination of the directionality of these relationships. Although our study identified a correlation between anxiety and depression scores, it did not identify the socio-demographic, student, and clinical predictors of anxiety and depression. External pressures, including personal life events or larger social issues, may have also contributed to students’ mental health difficulties. Our study may not have sufficiently accounted for these external effects, which can significantly impact students’ psychological well-being.

The overall response rate of 57% in our study, however moderate, is consistent with prior research examining mental health outcomes in medical students. Response rates in comparable populations often fluctuate between 40% and 60%, owing to the substantial academic demands and recurrent exposure to survey solicitations. Nonetheless, the possibility of non-response bias cannot be completely dismissed, as students suffering from significant anxiety or depression may have been either more or less inclined to engage. Nonetheless, the attained sample size was sufficient for statistical analysis and represents a diversified cohort for demographic and academic attributes, thereby reinforcing the findings’ representativeness. Regrettably, the absence of extensive demographic data on non-respondents precluded a systematic evaluation of non-response bias. Our study also did not examine early versus late response bias. We advocate for subsequent research to explore supplementary engagement tactics to enhance involvement and methodologies for assessing and mitigating these biases.

Although our study offers significant insights into the mental health of medical students, it is crucial to recognise its limits. Future research should focus on incorporating larger, more varied samples, employing longitudinal designs, and accounting for contextual and external factors to improve the understanding of the mental health difficulties encountered by medical students.

## 6. Conclusions

This study offers essential insights into the concerning prevalence of anxiety and depression among medical undergraduates at a rural South African university, indicating that 38% of participants displayed symptoms of GAD and 67.8% reported depressive symptoms. These rates stand significantly beyond global averages, highlighting the distinct psychosocial challenges encountered by medical students in rural settings. The study identified a strong association between mental health outcomes and critical variables, including the year of study, age, history of psychiatric illness, exposure to trauma, ethnicity, household structure, and academic difficulties, especially module repetition. These findings underscore the cumulative effect of personal, academic, and socio-economic stressors in intensifying mental health vulnerabilities within this population. The moderate positive correlation identified between anxiety and depression scores indicates a bidirectional relationship that could exacerbate psychological suffering if neglected.

The study findings highlight the urgent need for the introduction of tailormade mental health interventions in medical schools, especially in underserved rural areas. Emphasis must be placed on establishing comprehensive support systems that include early identification of at-risk students, incorporation of mental health literacy into the medical curriculum, and delivery of culturally competent psychological therapies. Moreover, measures to destigmatise mental health issues in medical education are crucial for creating an environment where students feel encouraged to seek assistance without fear of judgement or professional consequences.

## Figures and Tables

**Figure 1 healthcare-13-00649-f001:**
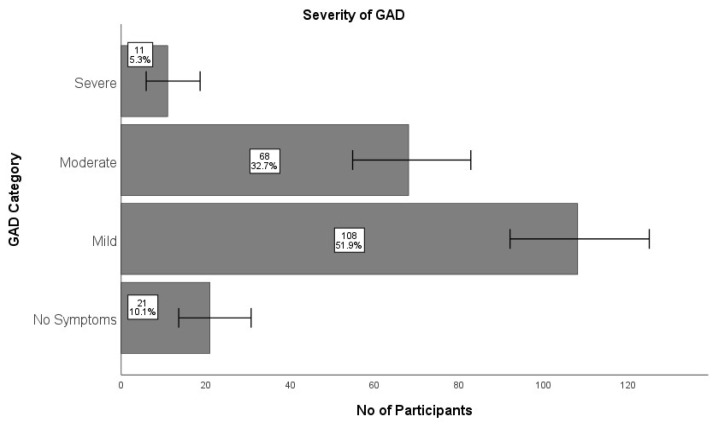
Severity of GAD among medical students (n = 208). Severe [95% CI: 6, 19]; Moderate [95% CI: 55, 83]; Mild [95% CI: 92, 125]; No Symptoms [95% CI: 14, 31]. CI: confidence interval.

**Figure 2 healthcare-13-00649-f002:**
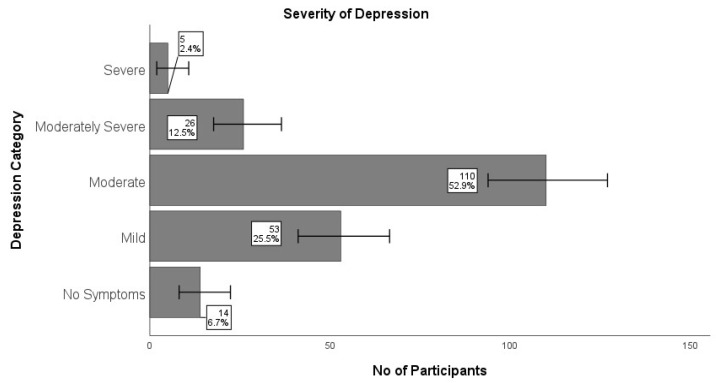
Severity of depression among medical students (n = 208). Severe [95% CI: 2, 11]; Moderately Severe [95% CI: 8, 37]; Moderate [95% CI: 94, 127]; Mild [95% CI: 41, 67]; No Symptoms [95% CI: 8, 22]. CI: confidence interval.

**Table 1 healthcare-13-00649-t001:** Socio-demographic, student, and clinical variables of the participants (n = 208).

Socio-Demographic, Student, and Clinical Variables		n (%)	*p*-Value *
Gender	Male	78 (37.5)	<0.001
	Female	130 (62.5)	
Ethnicity	Pedi	107 (51.4)	<0.001
	Tsonga	38 (18.3)	
	Venda	27 (13.0)	
	Others	36 (17.3)	
Age (in years)	18–19	56 (26.9)	<0.001
	20–21	61 (29.3)	
	22–23	37 (17.8)	
	>23	54 (26.0)	
Year of study	1	51 (24.5)	<0.001
	2	42 (20.2)	
	3	33 (15.9)	
	4	42 (20.2)	
	5	19 (9.1)	
	6	21 (10.1)	
Relationship	In a relationship	85 (40.9)	<0.001
	Single	123 (59.1)	
Religion	Christian	183 (88.0)	<0.001
	None/Others	25 (12.0)	
Receiving bursary?	Yes	183 (88.0)	<0.001
	No	25 (12.0)	
Monthly allowance (ZAR)	0–500	7 (3.4)	<0.001
	501–2000	129 (62.0)	
	>2000	72 (34.6)	
Did you fail any modules this year?	Yes	21 (10.1)	<0.001
	No	187 (89.9)	
Repeating modules this year?	Yes	16 (7.7)	<0.001
	No	192 (92.3)	
Accommodation	University residence	162 (77.9)	<0.001
	Private residence	46 (22.1)	
Do you have any children?	Yes	24 (11.5)	<0.001
	No	184 (88.5)	
Household income (South African Rands)	0–5000	24 (11.5)	<0.001
	5001–10,000	40 (19.2)	
	10,001–20,000	55 (26.4)	
	>20,000	89 (42.8)	
Family residence	Rural	155 (74.5)	<0.001
	Urban	53 (25.5)	
Household type (cohabitation)	Living with a single parent	102 (49.0)	<0.001
	Living with both parents	85 (40.9)	
	Living with relatives	21 (10.1)	
Prior exposure to trauma	Yes	64 (30.8)	<0.001
	No	144 (69.2)	
History of psychiatric illness	Yes	26 (12.5)	<0.001
	No	182 (87.5)	
Presence of other chronic illnesses	Yes	26 (12.5)	<0.001
	No	182 (87.5)	
Received professional psychological support before?	Yes	38 (18.3)	<0.001
	No	170 (81.7)	

* One-sample Kolmogorov–Smirnov (K–S) test.

**Table 2 healthcare-13-00649-t002:** Responses to the GAD-7 by medical students (n = 208).

GAD Question	Not at All (0) n (%)	Several Days (1) n (%)	Over Half the Days (2) n (%)	Nearly Every Day (3) n (%)	Mean (±SD)
1. Feeling nervous, anxious, or on edge.	43 (20.7)	88 (42.3)	63 (30.3)	14 (6.7)	1.23 (0.85)
2. Not being able to stop or control worrying.	19 (9.1)	120 (57.7)	52 (25.0)	17 (8.2)	1.32 (0.75)
3. Worrying too much about different things.	43 (20.7)	94 (45.2)	46 (22.1)	25 (12.0)	1.25 (0.92)
4. Trouble relaxing.	30 (14.4)	91 (43.8)	70 (33.7)	17 (8.2)	1.36 (0.83)
5. Being so restless that it’s hard to sit still.	56 (26.9)	85 (40.9)	49 (23.6)	18 (8.7)	1.14 (0.91)
6. Becoming easily annoyed or irritable.	54 (26.0)	78 (37.5)	59 (28.4)	17 (8.2)	1.19 (0.92)
7. Feeling afraid as if something awful might happen.	54 (26.0)	91 (43.8)	45 (21.6)	18 (8.7)	1.13 (0.90)

**Table 3 healthcare-13-00649-t003:** Responses to the PHQ-9 questionnaire by the medical students (n = 208).

PHQ Question	Not at All (0) n (%)	Several Days (1) n (%)	Over Half the Days (2) n (%)	Nearly Every Day (3) n (%)	Mean (±SD)
1. Little interest or pleasure in doing things.	37 (17.8)	112 (53.8)	41 (19.7)	18 (8.7)	1.19 (0.83)
2. Feeling down, depressed, or hopeless.	56 (26.9)	87 (41.8)	49 (23.6)	16 (7.7)	1.12 (0.90)
3. Trouble falling or staying asleep or sleeping too much.	26 (12.5)	76 (36.5)	73 (35.1)	33 (15.9)	1.54 (0.91)
4. Feeling tired or having little energy.	26 (12.5)	99 (47.6)	66 (31.7)	17 (8.2)	1.36 (0.80)
5. Poor appetite or overeating.	32 (15.4)	84 (40.4)	70 (33.7)	22 (10.6)	1.39 (0.87)
6. Feeling bad about yourself—or that you are a failure or have let yourself or your family down.	56 (26.9)	69 (33.2)	67 (32.2)	16 (7.7)	1.21 (0.93)
7. Trouble concentrating on things, such as reading the newspaper or watching television.	42 (20.2)	71 (34.1)	65 (31.3)	30 (14.4)	1.40 (0.97)
8. Moving or speaking so slowly that other people could have noticed. Or the opposite—being so fidgety or restless that you have been moving around a lot more than usual.	123 (59.1)	46 (22.1)	29 (13.9)	10 (4.8)	0.64 (0.90)
9. Thoughts that you would be better off dead, or of hurting yourself in some way.	42 (20.2)	108 (51.9)	42 (20.2)	16 (7.7)	1.15 (0.83)

**Table 4 healthcare-13-00649-t004:** Association between socio-demographic, student, and clinical variables with total GAD-7 and PHQ-9 scores (n = 208).

Socio-Demographic, Student, and Clinical Characteristics		Total GAD-7 Score (0–21)		Total PHQ-9 Score (0–27)	
		Mean (±SD)	*p*-Value	Mean (±SD)	*p*-Value
Gender	Male	8.69 (3.61)	0.63	10.51 (4.51)	0.09
	Female	8.54 (3.51)		11.31 (3.94)	
Ethnicity	Pedi	8.85 (3.27)	0.01 *	11.35 (3.91)	0.12
	Tsonga	7.34 (3.45)		10.08 (4.17)	
	Venda	10.07 (4.75)		11.96 (4.89)	
	Others	8.06 (2.89)		10.28 (4.25)	
Age (in years)	18–19	8.05 (3.14)	0.10	11.77 (3.01)	0.00 *
	20–21	8.33 (3.25)		11.90 (2.95)	
	22–23	8.30 (3.02)		9.89 (4.71)	
	>23	9.67 (4.35)		9.98 (5.50)	
Year of study	1	7.63 (2.90)	<0.001 *	11.16 (2.91)	<0.001 *
	2	8.76 (3.35)		12.71 (2.18)	
	3	8.58 (3.20)		12.21 (2.96)	
	4	7.38 (3.36)		6.98 (3.97)	
	5	12.26 (3.96)		14.95 (5.12)	
	6	9.76 (3.51)		9.86 (4.53)	
Relationship	In a relationship	8.51 (3.30)	0.77	10.38 (4.20)	0.10
	Single	8.66 (3.71)		11.45 (4.14)	
Religion	Christian	8.63 (3.47)	0.40	11.14 (4.22)	0.22
	None/Others	8.32 (4.10)		10.08 (3.76)	
Receiving bursary?	Yes	8.52 (3.60)	0.59	11.09 (4.03)	0.45
	No	8.87 (3.31)		10.71 (4.68)	
Monthly allowance (ZAR)	0–500	7.86 (3.53)	0.63	9.29 (5.77)	0.43
	501–2000	8.47 (3.65)		11.17 (4.06)	
	>2000	8.89 (3.37)		10.89 (4.22)	
Did you fail any modules this year?	Yes	7.95 (2.77)	0.47	12.29 (2.97)	0.10
	No	8.67 (3.61)		10.87 (4.27)	
Repeating modules this year	Yes	10.38 (3.96)	0.05 *	14.19 (2.64)	<0.001 *
	No	8.45 (3.47)		10.74 (4.17)	
Accommodation	University residence	8.75 (3.54)	0.22	11.12 (4.34)	0.60
	Private residence	8.04 (3.52)		10.63 (3.52)	
Do you have any children?	Yes	9.88 (4.01)	0.11	11.00 (5.40)	0.55
	No	8.43 (3.45)		11.01 (4.00)	
Household income (South African Rands)	0–5000	7.75 (4.86)	0.00 *	10.08 (4.76)	0.65
	5001–10,000	7.70 (3.70)		11.23 (4.14)	
	10,000–20,000	8.36 (3.12)		10.62 (4.28)	
	>20,000	9.37 (3.17)		11.40 (3.96)	
Family residence	Rural	8.56 (3.49)	0.91	10.80 (4.11)	0.25
	Urban	8.70 (3.70)		11.62 (4.32)	
Household type (cohabitation)	Living with a single parent	9.01 (3.84)	0.11	11.72 (4.40)	0.04 *
	Living with both parents	7.91 (3.01)		10.39 (3.72)	
	Living with relatives	9.38 (3.68)		10.10 (4.39)	
Prior exposure to trauma	Yes	9.61 (3.77)	0.01 *	11.73 (5.10)	0.12
	No	8.15 (3.35)		10.69 (3.66)	
History of psychiatric illness	Yes	11.15 (3.52)	<0.001 *	13.73 (5.27)	0.00 *
	No	8.23 (3.40)		10.62 (3.85)	
Presence of other chronic illnesses	Yes	8.42 (3.71)	0.57	10.81 (2.55)	0.77
	No	8.62 (3.52)		11.04 (4.36)	
Received professional psychological support before?	Yes	9.71 (4.32)	0.08	11.29 (5.26)	0.68
	No	8.35 (3.30)		10.95 (3.90)	

* *p*-value < 0.05 (Mann–Whitney U test and Kruskal–Wallis test where applicable); SD: Standard deviation.

**Table 5 healthcare-13-00649-t005:** Correlation between total GAD-7 and PHQ-9 scores reported by medical students (n = 208).

Mean total GAD-7 (±SD)	8.60 (3.54)
Mean total PHQ-9 (±SD)	11.01 (4.17)
Correlation coefficient (*r*_s_)	0.400
Degrees of freedom (*df*)	206
*p*-value	<0.001

## Data Availability

The authors will provide datasets generated and analysed during the current study upon request.

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
