# Peer review of "Exploring Anxiety and Depression Among Medical Undergraduates in South Africa: A Cross-Sectional Survey"

_healthcare, 2025, doi:10.3390/healthcare13060649_

Round 1
Reviewer 1 Report
Comments and Suggestions for Authors
This manuscript provides valuable insights into the prevalence of anxiety and depression among medical students in South Africa. The study addresses a critical issue in medical education, as mental health challenges can significantly impact students' academic performance and overall well-being. However, several concerns must be addressed to strengthen the study's clarity and rigor.
-
Significance of the study:
-
The introduction does not clearly articulate the devastating effects of increased anxiety and depression among medical students. A more detailed discussion on how these mental health challenges impact academic performance, clinical decision-making, patient care, and long-term career sustainability would enhance the manuscript’s impact.
-
-
Power Analysis:
-
Before conducting the survey, it is important to determine whether a power analysis was conducted to establish an adequate sample size. Clarifying this would help ensure that the study results are statistically meaningful and representative of the target population.
-
-
Data Analysis and Inferential Statistics:
-
The current data analysis lacks inferential statistics that could help identify potential predictors of high anxiety and depression among participants. Including regression analysis or other appropriate statistical methods would strengthen the manuscript by offering deeper insights into factors contributing to these mental health issues. This should be addressed as the study's limit in the "Discussion."
-
4. Ethical Concerns:
-
-
The current manuscript doesn't explain if this study has ethical approval, such as IRB approval.
-
Author Response
Comment 1: Significance of the study: The introduction does not clearly articulate the devastating effects of increased anxiety and depression among medical students. A more detailed discussion on how these mental health challenges impact academic performance, clinical decision-making, patient care, and long-term career sustainability would enhance the manuscript’s impact. |
Response 1: Thank you for pointing this out. We agree with this comment. Therefore, we have added more information about the impact of mental health challenges on academic performance, clinical decision making, patient care and career sustainability. Additional information pertaining to your comments with applicable references was added from Lines 79-97. |
Comment 2: Before conducting the survey, it is important to determine whether a power analysis was conducted to establish an adequate sample size. Clarifying this would help ensure that the study results are statistically meaningful and representative of the target population. Response 2: Thank you for this comment. We have consulted a biostatistician prior to the formulation of study objectives. Power analysis was conducted to ensure the sample was representative of the target population. The maximum sample required for this study considering the associations and correlations was 174. We have recruited 208 students in this study. We believe that the appropriate tests were used to ensure the study results are meaningful. No amendments were made in the manuscript. |
Comment 3: The current data analysis lacks inferential statistics that could help identify potential predictors of high anxiety and depression among participants. Including regression analysis or other appropriate statistical methods would strengthen the manuscript by offering deeper insights into factors contributing to these mental health issues. This should be addressed as the study's limit in the "Discussion." |
Response 3: Thank you for this comment. We completely agree with the suggestion made. We admit this as a limitation of our study, and we duly incorporated a statement in the limitation section of the manuscript found on lines 571-573.
Comment 4: Ethical Concerns: The current manuscript doesn't explain if this study has ethical approval, such as IRB approval. Response 4: Thank you for this comment. We have previously included this information in ‘’Institutional Review Board Statement: “The study was conducted in accordance with the Declaration of Helsinki and approved by the Turfloop Research Ethics Committee of the University of Limpopo (TREC/1494/2024:UG; 19 August 2024) for studies involving humans” 9 (Lines 609-611). |
Point 1: Quality of English language: The English is fine and does not require any improvement. |
Response 1: Thank you. |
5. Additional clarifications |
None. We thank the reviewer for providing valuable comments which certainly enhanced the quality of the manuscript. |

Reviewer 2 Report
Comments and Suggestions for Authors
- This study is a very simplistic approach that relates the level of depressive and anxiety symptoms to different variables in medical students. We know that these symptoms are important. I think it should be apparent from the abstract what the research gap is, i.e., where something is missing in the literature. This article comes to fill in the gap.
2. In lines 166-168 of the manuscript, you say, "The GAD-7 and PHQ-9 questionnaires exhibited strong test-retest reliability and good internal consistency. Both questionnaires were translated into multiple languages, extensively used in clinical practice and research globally". This statement, even with references from the literature, is insufficient. You should do your own internal consistency (Cronbach's Alpha) calculation for the group you studied to check whether the instrument behaved statistically correctly in your study. Figures 1 and 2 show a distribution of the totals, which means that you have summed the scores for each question into a total score. Reliability indicators become mandatory.
3. Moreover, testing the normality of the distribution of the scores for each quantitative variable is mandatory (use the One-Sample Kolmogorov-Smirnov test for this). If the scores do not distribute normally , you cannot do parametric tests such as ANOVA and T-student. However, you must look for their non-parametric equivalents (such as Kruskal-Wallis and Mann-Whitney).
4. You collect variables such as the participants' religion, but nowhere in the introduction do I find any reference in the literature that clarifies why religion would be relevant. I'm not saying that religion would not be relevant, but you need to show where you got the idea that religion might play a role. If you can't find any articles on this topic at all, at least provide philosophical/theological arguments for why you expect religion to influence these issues.
5. It is, indeed, very interesting that you found significant differences by ethnicity, year of study, income, repetition of study modules, cohabitation, and history of psychiatric illness. For each of these variables, you should ground your discourse in the literature (where you got the idea that these differences might occur).
6. Remember to include degrees of freedom when reporting correlation coefficients.
7. I congratulate you on the 'Discussion' part, as you understand well what this part of the article is about. However, you have not discussed the findings related to ethnicity, income, and repetition of study modules. You must introduce and discuss these issues; you cannot simply leave them presented. As with the other variables, show how what you have found differs from or resembles literature.
8. The conclusion does not do justice to your work. Please rewrite it so that the interesting findings in the article can be summarized.
9. This phrase: ”Anxiety and depression among medical students have emerged as significant public health concerns globally. Medical students are particularly vulnerable to these mental health challenges, which may have profound implications for their academic performance and future professional practice.” - is detected by my software as A.I. Generated
Author Response
Comment 1: This study is a very simplistic approach that relates the level of depressive and anxiety symptoms to different variables in medical students. We know that these symptoms are important. I think it should be apparent from the abstract what the research gap is, i.e., where something is missing in the literature. This article comes to fill in the gap. |
Response 1: Thank you for pointing this out. We agree with this comment. Therefore, we have added more information about the research gaps in the abstract and the introduction section. Additional information was added in the introduction section to further strengthen our case. Additional information pertaining to your comments was included in the abstract (Lines 27-31) and the introduction section (Lines 79-97) and (Lines 252-257). |
Comment 2: In lines 166-168 of the manuscript, you say, "The GAD-7 and PHQ-9 questionnaires exhibited strong test-retest reliability and good internal consistency. Both questionnaires were translated into multiple languages, extensively used in clinical practice and research globally". This statement, even with references from the literature, is insufficient. You should do your own internal consistency (Cronbach's Alpha) calculation for the group you studied to check whether the instrument behaved statistically correctly in your study. Figures 1 and 2 show a distribution of the totals, which means that you have summed the scores for each question into a total score. Reliability indicators become mandatory. Response 2: Thank you for this comment. We have conducted reliability tests to verify if the tools used in our study had good internal consistency. The results of the internal consistency were presented in Results section, Section 3.2. Overall, both the tools had an acceptable level of internal consistency (0.67 for both). Addressed in Lines 353-361. |
Comment 3: Moreover, testing the normality of the distribution of the scores for each quantitative variable is mandatory (use the One-Sample Kolmogorov-Smirnov test for this). If the scores do not distribute normally, you cannot do parametric tests such as ANOVA and T-student. However, you must look for their non-parametric equivalents (such as Kruskal-Wallis and Mann-Whitney) |
Response 3: Thank you for this comment. We completely agree with the suggestion made. We have reviewed our data analysis methods and tests. Normality tests for socio-demographic, student and clinical characteristics were conducted with One-Sample Kolmogorov-Smirnov test which revealed that our data is non-parametric. Therefore, we have utilised Mann-Whitney U test (for Independent Samples t-test) and Kruskal Wallis H test (in place of ANOVA). P-values were amended and results were amended based on the results. Amendments could be found on Lines 328-329, 337-338, 352, 410. Results section and tables were modified accordingly.
Comment 4: You collect variables such as the participants' religion, but nowhere in the introduction do I find any reference in the literature that clarifies why religion would be relevant. I'm not saying that religion would not be relevant, but you need to show where you got the idea that religion might play a role. If you can't find any articles on this topic at all, at least provide philosophical/theological arguments for why you expect religion to influence these issues. Response 4: Thank you for this comment. We have included a paragraph on the relevance of religion in our study with applicable references (Lines 202-209). |
Comment 5: It is, indeed, very interesting that you found significant differences by ethnicity, year of study, income, repetition of study modules, cohabitation, and history of psychiatric illness. For each of these variables, you should ground your discourse in the literature (where you got the idea that these differences might occur). Response 5: Thank you for this comment and we fully agree with the recommendation. We have provided detailed explanation of each of the socio-demographic, student and clinical characteristics supported by the literature in the introduction section. (Lines 135-247) which solidified the introduction section and also partially addressed the Comment 1.
Comment 6: Remember to include degrees of freedom when reporting correlation coefficients Response 6: Thank you for this valuable comment. We have included degrees of freedom in our reporting in the abstract, results section and Table 5 (Lines 51 and 412).
Comment 7: I congratulate you on the 'Discussion' part, as you understand well what this part of the article is about. However, you have not discussed the findings related to ethnicity, income, and repetition of study modules. You must introduce and discuss these issues; you cannot simply leave them presented. As with the other variables, show how what you have found differs from or resembles literature. Response 7: Thank you for this valuable comment. Ater the data analysis methods were modified; we found significant differences mean scores in ethnicity (for anxiety only), income (anxiety) and repetition of module (both anxiety and depression). We have addressed these aspects in the discussion section with the applicable legislation (Lines 515-539).
Comment 8: The conclusion does not do justice to your work. Please rewrite it so that the interesting findings in the article can be summarized. Response 8: Thank you for this comment. We have rewritten the conclusion section and incorporated key findings from our study and included several recommendations based on our study findings (Lines 583-599).
Comment 9: This phrase: ”Anxiety and depression among medical students have emerged as significant public health concerns globally. Medical students are particularly vulnerable to these mental health challenges, which may have profound implications for their academic performance and future professional practice.” - is detected by my software as A.I. Generatede Response 9: Thank you for this comment and we apologize for this error. We have replaced this sentence in the abstract section (Lines: 23-37).
Point 1: Quality of English language: The English is fine and does not require any improvement. |
Response 1: Thank you. |

Reviewer 3 Report
Comments and Suggestions for Authors
Thank you for allowing me to review your important work. The article is very well written. I would suggest that you include the word 'quantitative" in the title to alert readers to your methodology when searching. Suggestion: "Under Pressure: Quantitatively Exploring....."
Line 166 - Can you include psychometric statistics with the description of the tools? Such as a Cronbach's alpha to show the readers the reliability of the tools in addition to you stating that they are reliable. Would give more credence to your statements.
line 259 - You state that the: Students aged 18-19 exhibited heightened level of depression (11.90) but your table assigns this data point the 20-21 age group. Please correct.
Discussion section needs a limitations section.
Otherwise, I am very pleased with the article and look forward to seeing it published.
Author Response
Comment 1: Thank you for allowing me to review your important work. The article is very well written. I would suggest that you include the word 'quantitative" in the title to alert readers to your methodology when searching. Suggestion: "Under Pressure: Quantitatively Exploring....." |
Response 1: Thank you for pointing this out. We agree with this suggestion. Therefore, we have amended the title accordingly (Line 2). |
Comment 2: Line 166 - Can you include psychometric statistics with the description of the tools? Such as a Cronbach's alpha to show the readers the reliability of the tools in addition to you stating that they are reliable. Would give more credence to your statements. Response 2: Thank you for this comment. We have included additional information about the internal consistency about the tools used in this study with supporting literature (Lines: 298-303). Moreover, we also conducted reliability tests to verify if the tools used in our study had good internal consistency. The results of the internal consistency were presented in Results section, Section 3.2. Overall, both the tools had an acceptable level of internal consistency (0.67 for both). Addressed in Lines 353-361. |
Comment 3: Line 259 - You state that the: Students aged 18-19 exhibited heightened level of depression (11.90) but your table assigns this data point the 20-21 age group. |
Response 3: Thank you for this comment and notifying us about this error. We have corrected the statement (Lines 401-401).
Comment 4: Discussion section needs a limitations section. Response 4: Thank you for this comment and alerting us about this section. We have included a Limitations section addressing all the aspects (Lines 560-581). |
Point 1: Quality of English language: The English is fine and does not require any improvement. |
Response 1: Thank you. |

Round 2
Reviewer 2 Report
Comments and Suggestions for Authors
I agree to publish the manuscript after the latest improvements.
Author Response
Are the conclusions supported by the results? |
Can be improved |
Thank you. The Conclusion section was fully amended, and all the significant results were added. Addressed from lines 694 to 713/ |